# Exploring the Cell Stemness and the Complexity of the Adipose Tissue Niche

**DOI:** 10.3390/biom11121906

**Published:** 2021-12-19

**Authors:** Nadav Kislev, Roza Izgilov, Raizel Adler, Dafna Benayahu

**Affiliations:** Department of Cell and Developmental Biology, Sackler School of Medicine, Tel Aviv University, Tel Aviv 6997801, Israel; nadavkvdt@gmail.com (N.K.); roza21854@gmail.com (R.I.); raizelsober@gmail.com (R.A.)

**Keywords:** mesenchymal cells, immune cells, adipose tissue, extracellular matrix

## Abstract

Adipose tissue is a complex organ composed of different cellular populations, including mesenchymal stem and progenitor cells, adipocytes, and immune cells such as macrophages and lymphocytes. These cellular populations alter dynamically during aging or as a response to pathophysiology such as obesity. Changes in the various inflammatory cells are associated with metabolic complications and the development of insulin resistance, indicating that immune cells crosstalk with the adipocytes. Therefore, a study of the cell populations in the adipose tissue and the extracellular matrix maintaining the tissue niche is important for the knowledge on the regulatory state of the organ. We used a combination of methods to study various parameters to identify the composition of the resident cells in the adipose tissue and evaluate their profile. We analyzed the tissue structure and cells based on histology, immune fluorescence staining, and flow cytometry of cells present in the tissue in vivo and these markers’ expression in vitro. Any shift in cells’ composition influences self-renewal of the mesenchymal progenitors, and other cells affect the functionality of adipogenesis.

## 1. Introduction

The motivation to study adipose tissue (AT) is related to the functional role played by the organ in lipid metabolism and as a source of endocrine factors [1,2]. Two main types of adipose tissue, white adipose tissue (WAT) and brown adipose tissue (BAT), control energy homeostasis. Although these subtypes share a mesenchymal origin, they differ in their distribution, function, and morphology, where BAT cells contain multiple small lipid droplets, and WAT cells have a single lipid droplet [3]. Another difference is that BAT precursor cells express the Myf5 and Pax7 transcription factors, while WAT precursor cells are Myf5-negative [4,5,6]. The different types of AT are also distributed in anatomically distinct locations, where BAT is found mainly around the shoulders and ribs and is most common in infants and in adults who have adapted to cold. In contrast, WAT is more widely distributed and can be subdivided into subcutaneous fat (SAT) and visceral adipose tissue (VAT) [3]. The SAT accounts for 80% of body fat and is located underneath the skin throughout the whole body, and VAT constitutes the rest of body fat and is mainly in the abdominal cavity [7]. The WAT plays a major role in body physiology and serves as an energy depot, which places the VAT in a position to control the balance between lipolysis and lipogenesis [8]. In addition, WAT secretes leptins, adiponectin and other peptides that affect the body’s food intake and energy expenditure, thereby functioning as an endocrine organ [1,2]. In this context, VAT is recognized as being closely associated with insulin resistance and metabolic syndrome [9,10,11,12,13].

Despite intense research, the regulatory events guiding progenitor cells activation and differentiation in WAT remain largely unknown. The adipose tissue comprises a complex assortment of different cellular populations, including adipocytes, mesenchymal stem and progenitor cells, and immune cells, such as macrophages and lymphocytes [3,14,15]. These cell populations have continuous interactions also seen when adipocytes undergo apoptosis, and they are surrounded by a crown-like structure of M2 macrophages that are responsible for their clearance [16,17]. Similarly, events associated with chronic inflammation also affect proliferation and maturation of pre-adipocytes or adipocytes [18,19,20,21]. The recruitment and differentiation of mesenchymal progenitors may be altered in response to physical injury or nutritional effects such as hyperglycemia or a high-fat diet [17,22]. Such conditions also alter the matrix composition and influence adipocyte cell fate with respect to their metabolism, as well as directing the polarization, fate, and activation of macrophages [14,15,23,24,25].

### How Can We Study Adipose Derived Stem Cells (ADSCs) in the Tissue Niche?

Adipose tissue exhibits extensive plasticity in response to environmental stimuli and nutritional status [26]. A variety of microscope and staining techniques were developed to study the cells and extracellular matrix (ECM) components that affect the physiology and biology of adipose tissue [27,28,29]. While conventional 2D visualization methods are informative, they provide limited opportunities to study the tissue as a whole, and in particular, fail to consider the 3D architecture of the organ. A combination of methods is used to study various parameters of cells and ECM and investigate the factors that determine the fate of cells; the resident mesenchymal and immune cells or other infiltrating immune cells. Interrelated in vitro models are applied to explore the different subpopulations of cells existing in the adipose tissue [30,31,32,33,34].

The current study presents methods that allow dissecting the VAT structure by utilizing a variety of imaging techniques and biochemical analyses to identify specific proteins, related to different resident cells in the adipose tissue, as well as on isolated cell populations (Figure 1). Combinations of methods provide valuable information about VAT adipose tissue structure and aid in understanding the tissue function and the specific types of cells responding to different stimuli. The complementary nature of the methods and analyses counteract the limitations of each individual approach, and the final combination provide useful tools for identifying cell populations as well as the tissue structure. The results presented here provide valuable information about the types of cells harnessed to combat the metabolic pathology and the potential recognition of progenitor cells in tissues that play a role also in regeneration processes.

## 2. Material and Methods

Visceral adipose tissues (VAT): Epididymal visceral adipose tissue were collected from C57BL/6J mice and immediately used as fresh, frozen (by liquid nitrogen), and fixed tissues for further procedures, as detailed below and presented in the schematic illustration (Figure 1). The mice were kept in a conventional facility with 12 h light/dark cycles and were fed with standard chow and provided water ad libitum. Animal care and experiments were in accordance with the guidelines of the IACUC Approval (01-21-044). Each experiment was performed on 3–5 samples.

Whole-Mount and immunofluorescence staining and live imaging: VAT whole-mount staining was performed as previously described [29]. The samples were stained with primary antibodies; Glucose transporter 4 (GLUT4; SC 53566), Perilipin 1 (PLIN1; Sc-390169), both from Santa Cruz. CD45 and CD34 (Bio-gems, Westlake Village, CA, USA) and secondary antibodies, Cy3 -anti-mouse and 647- anti-rat (Jackson). Lipid droplets content was stained with 10 µg/mL Nile Red (Sigma N-3013) dye. Live imaging of freshly isolated tissues incubated with Cholera Toxin Subunit B (CTxB-647; Thermo Fisher, C34778) binding to ganglioside GM1 receptor was performed. Then, tissues were fixed with 4% formaldehyde containing 0.03M sucrose in PBS for 10 min at room temperature (RT). Before visualization, a fluoroshield mounting medium containing 4′, 6-diamidino-2-phenylindole (DAPI) (Electron Microscopy Sciences, #17985-10) was added to the tissues, then they were viewed and photographed by a confocal SP8 microscope (Leica, Germany).

Histology analysis: VAT was harvested and immediately fixed in 4% paraformaldehyde overnight at 4 °C, washed with PBS, ethanol, and xylene, then embedded into paraffin blocks to prepare five µm thick sections. For staining, sections were deparaffinized, rehydrated, and stained with Masson’s Trichrome (Sigma-Aldrich, St. Louis, MI, USA). Stained sections were photographed under Aperio slide scanner microscope (×200) used for imaging analysis. The images were processed and analyzed using the ImageJ software (NIH, Bethesda, MD). Single cells’ adipocyte area and the ECM fraction in the tissue were analyzed on various fields of view (FOV). Images were converted into a binary format to calculate the percentage of ECM in each FOV.

Transmission Electron Microscopy (TEM): VAT was fixed overnight in 2.5% Glutaraldehyde in phosphate-buffered (PBS) at 4 °C was then washed several times with PBS and post-fixed in 1% OsO4 in PBS for 2 h at 4 °C. Dehydration was carried out in graded ethanol and embedded for preparation of thin sections were mounted on Formvar/Carbon coated grids. Sections were stained with uranyl acetate and lead citrate, and examined using a JEM 1400 Plus transmission electron microscope (Joel, Japan). Images were captured using SIS Mega view III and the TEM imaging platform (Olympus).

Tissue extracellular matrix (ECM) and decellularization: VAT was decellularized according to the Wang protocol [35]. Briefly, three freeze and thaw cycles were performed, followed by an extensive wash in double-distilled water (DDW) for 2 days at RT with agitation at 120 rpm. The tissue was transferred to 0.5 M NaCl and 1 M NaCl for 4 h each and washed with DDW overnight. Next, the tissue was digested by 0.25% trypsin-EDTA washed and incubated with isopropanol overnight. Afterward, it was treated with 1% Triton X-100 for 3 days (changed daily), rinsed with PBS, and then stored in isopropanol at 4 °C or lyophilized. The resulting ECM was visualized unstained with polarized microscopy (Nikon Optiphot-2) and confocal microscopy (ex. 488 nm, em. 545 nm) (Zeiss 710).

Isolated cells from VAT: Mature adipocytes and the stromal vascular fraction (SVF) were isolated from VAT as described [36]. Briefly, the tissue was minced to a fine consistency in HBSS solution (Biological Industries, Israel), then incubated with collagenase solution (Sigma-Aldrich, c-5138) for one hour at 37 °C with shaking and the digested tissue was filtered through a 100 µm cell strainer (SPL life science) and centrifuged at 1800 rpm for 5 min. The mature adipocytes were collected for staining, and the pellet was suspended in red blood cells lysis buffer for 5 min, centrifuged at 1800 rpm for 5 min to collect the pellet containing the SVF cells. The isolated cells were stained with antibodies and analyzed by flow cytometry or cultured. For culturing purposes, the cells were suspended in GM containing Dulbecco’s modified Eagle’s medium (DMEM; Gibco) supplemented with 10% fetal bovine serum (Biological Industries), 1% L-glutamine (Biological Industries), 1% penicillin-streptomycin (Pen-Strep; Sigma), and 0.5% 4-(2-hydroxyethyl)-1-piperazine-ethanesulfonic acid (HEPES; Biological Industries, Israel), and seeded, the medium was replaced 24 h later to remove debris and other cell types. The adherent cells were grown to 90% confluency and then transferred into a differentiation medium containing the GM, supplemented with 5 μg/mL insulin (Sigma), 1 μM dexamethasone (Sigma), 400 μM 3-isobutyl-1-methylxanthine (IBMX; Sigma). Forty-eight hours later, the medium was changed to a maintenance medium consisting of the GM with insulin 5 μg/mL and replaced twice a week. The differentiating cells were monitored and analyzed for level of adipogenesis (LOA) (EVOS FL Auto 2, Invitrogen) as was described by our group [37].

Flow cytometry analysis: The isolated SVF cells were suspended in a staining buffer (PBS with 2% serum and 0.1% NaN3) for 15 min on ice. 10 [6] cells/sample were incubated with each marker; Anti-CD45-APC, Anti-CD34-FITC, Anti-CD31-Cy7 (Bio-gems, Westlake Village, CA, USA) and analyzed by Cytoflex5l and Kaluza software (Beckman Coulter instrument).

Biochemistry and protein identification by Dot blot (DB): Protein extraction from VAT [38] and samples analyzed by dot-blot (applied 2 μg of samples) on a nitrocellulose membrane and incubated overnight with primary antibodies; Plin1 (SC-390169), PPARγ (SC-7273), actin (MP-691001/2), and LPL (SC-373759) (all from Santa Cruz, USA), followed by incubation with goat anti-mouse horseradish peroxidase (HRP) conjugated (Jackson 115035) secondary antibody, the signal was detected with chemiluminescent substrate (Super Signal™ West Pico PLUS kit, 34578, Thermo Fisher, USA), exposed, and quantified digitally on Fusion FX7 (Vilber Lourmat).

Statistical analysis: Statistical analysis was performed using two-tailed, unpaired, *t*-test to identify significant differences in the samples. For each parameter, it was presented as the means and standard error and *p* < 0.05 was considered statistically significant. Distribution charts and plot were performed using the Graph Pad Prism 9 software.

Schematic illustrations were created by the Bio Render software https://biorender.com accessed on 12 December 2021.

## 3. Results

### Adipose Tissue Structure and the Main Subpopulations of Cells

Adipose tissue morphology can serve as a read-out of the nutrient status of the body. Adipose tissue is composed of numerous cell subpopulations and their interactions have an important impact on the tissue structure and function. We used a variety of complementary techniques to visualize the 2D and 3D morphology of the tissue architecture and to examine the cellular subpopulations and ECM components.

Whole-mount staining of ex-vivo-isolated tissue maintains the 3D structure and allows visualizing the native morphology of the organ with minimal processing steps. The tissue was stained using three markers of different cell compartments: the plasma membrane stained with an antibody for GLUT4, lipid droplet (LD) membrane stained by an antibody for Plin1, and the lipid content stained with the Nile-Red dye (Figure 2A). Adipocyte areas quantification, measured on a single cell basis, showed a range of 1000–13,000 μm [2] in cell size (Figure 2B). Adipocyte size is a gold standard parameter in common use in the context of nutrition and investigation as a read out of the metabolic state of the tissue. Whole-mount staining based quantification of adipocyte size allows measuring the cells in their native form. In addition, we employed live cell imaging, using the CTxB fluorescent probe that binds to ganglioside M1 (GM1) on the adipocyte membrane. VAT was incubated with CTxB for 5 or 30 min to allow probe binding to the membranes, followed by endocytosis, was monitored, and cells fluorescence increase with time was evident after 30 min (Figure 2D), demonstrating the ability to perform various live assays ex-vivo tissues.

Identifying adipocytes and other cell types in the tissue with specific markers make it possible to trace cell lineages and monitor cell fate. The whole-mount (WM) of VAT immunostaining allows visualizing and quantifying the presence and number of cells in the tissue. Antibodies to CD45, a marker for immune cells, identified positive cells located primarily at the junctions between adipocytes. As shown in Figure 3, CD45^+^ cells were present with an average of more than twenty cells per analyzed FOV, emphasizing the vast communication between the immune cells and adipocytes in the tissue. CD34^+^ fibroblast cells demonstrated in the tissue, with an average of seven cells per FOV. These cells considered stem cells or progenitors have a major role in the differentiation and regulation of the tissue. The WM method enables the detection of non-adipocytes in their niche, contributing to our understanding of the microenvironment.

The adipose tissue is composed of heterogeneous cellular subpopulations that can be isolated following digestion of the tissue ECM to “release” cells, as shown in the schematic illustration (Figure 4A). The two main isolated fractions represent the mature adipocytes in the supernatant, and the stromal vascular fraction (SVF) cells. The SVF contains heterogeneous non-adipocyte cells which were further subdivided by flow cytometry into mesenchymal progenitors, immune, and endothelial cells (Figure 4B). Positive cells for CD45 differentiate between hematopoietic and non-hematopoietic populations. The CD45^+^ cells were analyzed based on co-expression of CD31 or CD34. CD34 marker is expressed on both mesenchymal and endothelial populations, whereas CD31 expressed only on endothelial cells (Figure 4B). These markers allow for the characterization and separation of the populations of cells present in the SVF.

The adipose tissue is separated into mature adipocytes and the SVF cells. The mature adipocytes have a uni-lobular lipid droplet stained with Nile red and the cell nuclei in the periphery as presented by a nucleus staining (Figure 4C). The mesenchymal progenitors when cultured can differentiate into several lineages depending on the differentiation medium used. Here, the mesenchymal progenitor cells were isolated from the SVF cultured under differentiation medium into adipocytes with a follow up for their level of adipogenesis (LOA, Figure 4D). We visualized the cultures throughout their differentiation at 4, 11, and 15 days post adipogenesis induction possess an increasing numbers of adipocytes with time. The differentiation capacity of the cultures was quantified by LOA from macro imaging (×40) and the adipogenesis seen on high magnification images (×200) (Figure 4D). The isolation and identification of the stem cells and cellular sub populations in the tissue are important to identify as are the key functional cells in adipose tissue physiology.

The preparation of paraffin-embedded VAT enables a 2D visualization of stained tissue sections by light or fluorescence microscopes. The current analysis of adipocyte cells size and the ECM serves as an indicator of the nutritional status, together with the ECM indicate the remodeling and physiological status of the tissue. Here, the measured size of adipocytes on the paraffin tissue section of the “empty” space left after the lipid droplets were extracted during sample preparation. The results of the single-cell analysis demonstrated a cell size varying from 2000 µm [2] to 12,000 µm [2] (Figure 5A). Complementary analysis using the binary intensity area of images to evaluate the ECM in the tissue sections resulted with an average of 21% ECM of the cover area (Figure 5A).

To visualize VAT structure at higher resolution, we used the transmission electron microscope (TEM) (Figure 5B). Several mature adipocytes with a single lipid droplet occupy the cells are seen with an elongated nucleus. An undifferentiated precursor (marker in the box) located between the adipocytes, this cell nucleus is rounder and in the cells, we clearly see the endoplasmic reticulum (Figure 5B). Figure 5C demonstrates proteins expressed that were isolated from adipose tissue and analyzed by biochemical techniques using the dot-blot (DB). These methods detected the expression of number of adipose related proteins; PPARγ, a transcription regulator of adipocyte differentiation, the lipid droplet associated proteins; LPL, Plin1, and actin. The biochemical techniques enable us to quantify the expression levels of specific proteins of interest in the tissue.

The ECM is composed of a variety of proteins, including collagens and polysaccharides, provides a microenvironment for the cells. The ECM play a role on cells survival, influence their adhesion and shape, differentiation, and metabolism. In turn, as the cells differentiate, the composition of the ECM changes. This prompts the next step in the analysis, which employed various decellularization steps to analyze the ECM isolated from fresh adipose tissue (Figure 6). The tissue includes various cells (adipocytes, immune and endothelial) with the surrounding ECM. Sequential decellularization stages were used to remove cellular components while retaining the ECM of the tissue (Figure 6A). At first, fresh tissue has a yellowish/pinkish color derived from the lipid content and the red blood cells. After the decellularization stages, the tissue volume decreases substantially due to the release of the lipid content, and the specific gravity increases resulting in a white porous material. The isolated ECM is then stored in isopropanol or as a lyophilized matrix. The resultant ECM was observed unstained via microscopy. Figure 6B shows the ECM under light microscopy with or without a polarization filter to observe the organized collagen fibrils based on the collagen birefringence properties. While the unstained ECM is almost invisible, the use of polarized light revealed a prominent fibrillary structure based on the birefringent properties of the collagen fibrils (Figure 6B). The images are translated to a thermal range as depicted in the lower panel that highlight the structures detected as a heat map given to target envision where the profile verifies the protein identification. In addition, the collagen fibrils visualized by fluorescence microscopy detected the auto fluorescence as fibers, branching into thinner fibrils (Figure 6B). The remaining ECM has no clear tissue structure as the cellular content was removed during the decellularization.

## 4. Discussion

Fat tissue responds directly to nutritional stimuli, and the increase in adipose tissue mass leading to obesity is currently a major public health challenge. Obesity has an immediate effect on the body’s metabolism, and the consequences of this state have led to a pandemic of insulin resistance, dyslipidemia, and hypertension, among other complications [14,39]. While the role of adipocytes in energy storage has long been recognized, the notions of adipose tissue as a main immune and endocrine organ were established in the last decade [1,2,40].

In obesity, the adipose tissue expands both by hypertrophy of existing adipocytes and by hyperplasia, the formation of new adipocytes through differentiation of pre-adipocytes [41,42,43]. The new adipocytes that emerge from the differentiation of pre-adipocytes contribute to tissue expansion. The pre-adipocytes are fibroblast-like cells located in the perivascular that serve as the tissue stem cells [44,45,46]. Herein, we provided a series of methods to follow the intact adipose tissue and analyzed the mesenchymal and immune cells subpopulations. In addition to the cells, the tissue niche contains the ECM that provides signals to the resident or transient cells. The ECM is altered by physiological changes, and the various cells are modified according to the tissue function [25,33,47]. The identification of cell subpopulations is crucial for better understanding adipose tissue functionality. Moreover, knowing the state of the stem cells is essential for their role in tissue repair and is widely used in the approach of tissue engineering [48,49,50].

The use of imaging for 2D and 3D visualization of tissue architecture presented in this study allowed us to investigate the morphology and biology of adipose tissue. Visualization of both the cellular and ECM components of adipose tissue in whole-mount (WM) samples preserve the 3D adipose tissue morphology. However, although the WM-3D preparation has the advantage of being less destructive, the technique has some limitations in providing accurate information regarding deeper parts of adipose tissue and thus is complemented with 2D histology sections. Both techniques are complementarily used for tissue staining and identification of specific resident mesenchymal and immune cells together with other transient cells as well as ECM proteins. Interestingly, such histology of VAT adipocytes in vivo reveals a uni-lobular morphology with the nucleus in the periphery of the cells, which differs from the appearance of cell culture-derived adipocytes that are multi lipid droplets in the cells. Nevertheless, culture-derived adipocytes enable us to study multiple functional and molecular aspects that complement the morphological studies with ex-vivo and in vitro assays on cultured cells.

It is of prime importance to recognize the heterogeneous mesenchymal progenitor cell populations in the adipose tissue that serves as the reservoir responsible for tissue repair and have wide use in tissue regeneration [17]. Thus, the ability to isolate and culture the cells and monitor their differentiation into various cell lineages according to their environmental cues is of great interest. Since the focus of the current study relates to adipocytes, this dictated the differentiation medium used and the morphological characteristics of interest.

The mesenchymal progenitors were visualized in the interstitial space (Figure 3A arrows) in the whole-mount analysis, where they displayed a fibroblast morphology (with extended cell processes). These cells were also characterized by flow cytometry analysis as CD34+CD45-CD31-cells (Figure 4B). This technique is widely used in adipose tissue for the selection and isolation of mesenchymal subpopulations. Mesenchymal markers, such as CD105, CD90, and CD73, characterize and verify general mesenchymal subpopulations in tissues. It was shown in the literature that these markers are highly correlated with our selection approach in SVF-related studies. The CD34+CD31-gating approach is common in adipose tissue studies and is considered a reliable technique for isolating the general adipose-derived stem and progenitor subpopulations [51,52,53]. The SVF isolated cells were also cultured to follow adipogenic differentiation and measured for their level of adipogenesis (LOA, Figure 4D). This result confirms the identity of the cells and demonstrates their ability to become adipocytes are analyzed at the single-cell level. The approach also allows for tracking of other lineage fate under appropriate culture conditions to various cellular populations.

A transmission electron microscope enables the observation and characterization of subcellular structures and the identification of multicellular interactions in the cells’ niche that cannot be visualized by other methods (Figure 5). The study used multiple approaches (microscopy, flow cytometry, and biochemistry) to observe the location of specific proteins and examine their appearance in the tissue. It also demonstrates the ability to quantify specific proteins by employing biochemical assays that detect proteins’ presence and measure the amounts in adipose tissue. These effects are prominent in the ECM, but in addition, there may also be changes in the cellular populations inhabiting or passing transiently through the tissue in response to the physiological state of the body.

Decellularization of the adipose tissue enables us to study the tissue niche of the cells as the ECM is a key component in the tissue structure. The ECM contains a variety of proteins, and we analyzed the ECM following decellularization by imaging. The most notable protein in the ECM is collagen that contributes to auto fluorescence; an intrinsic property of the molecules [54]. Auto-fluorescence is the natural emission of light by biological structures under absorbed light. Thus, alterations in auto fluorescence are used to investigate the structure of interest or serve as a useful diagnostic indicator of change that is related to certain pathologies even in the absence of specific staining are benefit use of some optical imaging systems [55]. In particular, the architecture and behavior of collagen fibers are dynamic and reflect tissue function [56,57,58]. The arrangement of collagenous tissues relies on the viscoelastic properties resulting from the structural configuration and constituents of the fibrils [59]. On the microscale, collagen fibers are sinusoidal crimped, providing the straightening and re-orientation as a tensile load is applied on the tissue to provide the 3D structure and collagen fiber [59,60]. Polarization microscopy images of ECM fibers can serve as a read-out of the tissue physiology. Visualization of the fibers by polarized light microscopy provides the parameters for length and alignment of the helical fibers. The direction of coiling and supercoiling influence the collagen organization and the intensity of the birefringence. Increased birefringence is caused by stretching attributes to the intermolecular changes rather than intramolecular alterations [61].

The collagen fiber’s structure is changed during pathology or aging. It is also recognized that not all collagens have equal birefringence properties; this is mainly influenced by the large side chains and amounts of interstitial proteoglycans. For example, the composition of collagen III found in healing wounds differs from other forms and is rich in intermolecular carbohydrate components [62]. Birefringence, visualized using the polarizing filters, is already commonly utilized in medical diagnostic, and the bright birefringence result against a dark background has been used in other tissues, such as in ophthalmology or dermatology to view skin lesions [63,64,65]. This illustrates how the biophysical properties of the ECM can be useful in studying the ECM composition, stiffness, porosity, and fiber alignment anisotropy that affect tissue architectures and play a crucial role in controlling cell behavior in vivo.

## 5. Perspective

This study aimed to provide a better understanding of the physiological and pathological parameters of adipose tissue. It offers a set of tools for investigating how remodeling of the organ plays a role in regulating whole-body energy metabolism in response to the nutritional status and crosstalk between adipocytes and SVF cells. Multiple cellular responses are dynamically altered during adipose tissue expansion as well as the ECM remodeling and angiogenesis.

Understanding the features of the various adipose depots is crucial to the recognition and modulation of the resident adipose stem cells (ASCs). The ASCs and SVFs are important for regenerative medicine as they have the ability to differentiate into multiple cell types. The cell fate depends on a combination of factors that may be present in the cell niche, and this, in turn, influences the tissue physiology and the present cell subpopulations. In particular, mature adipocytes play an essential metabolic regulatory function, and therefore, identifying factors regulating the ASCs in situ in the cells’ niche and developing methods to follow them in vivo is of prime importance.

## Figures and Tables

**Figure 1 biomolecules-11-01906-f001:**
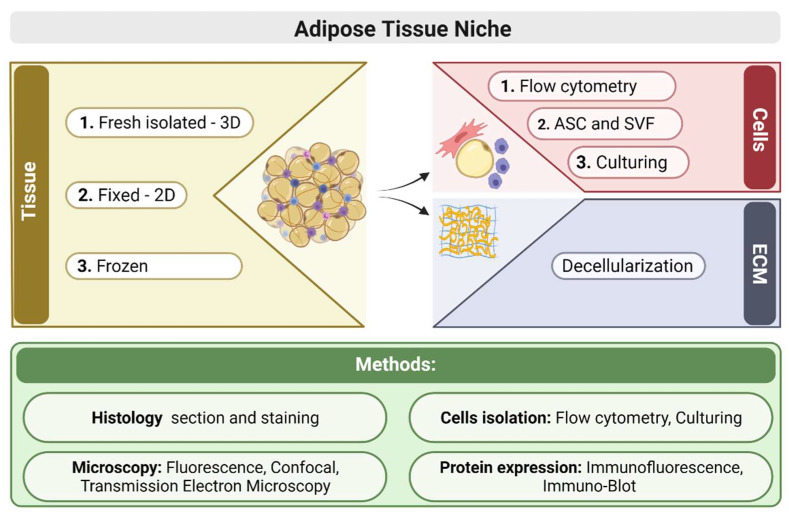
Schematic scheme for methods employed in the study of adipose tissue and cells.

**Figure 2 biomolecules-11-01906-f002:**
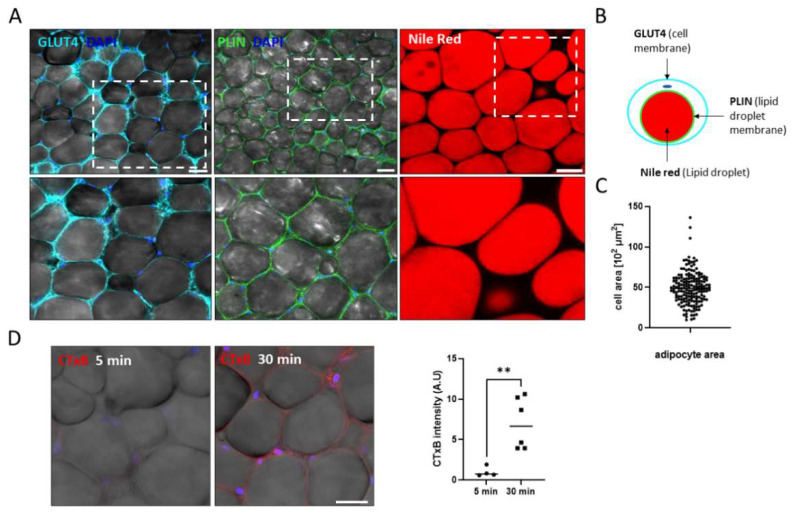
Whole-mount 3D tissue staining of different cellular compartments. (**A**) Whole-tissue staining for GLUT4 (membrane), Plin1 (lipid droplets membrane), and Nile-Red (lipid droplets content) (scale bar = 50 μm) also depicted in (**B**) schematic illustration of different cellular compartment staining. (**C**) Adipocyte’s area measurement distribution (n = 175). (**D**) Ex-vivo binding of CTxB fluorescent probe after 5 and 30 min. Nucleus staining-DAPI, scale bar = 25 μm, and a CTxB signal intensity analysis after 5 and 30 min incubation; data were normalized to the 5 min expression (n = 4 and 6, respectively), significance was calculated using a two-tailed unpaired Student’s *t*-test. ** *p* < 0.01. Error bars represent means ± SD.

**Figure 3 biomolecules-11-01906-f003:**
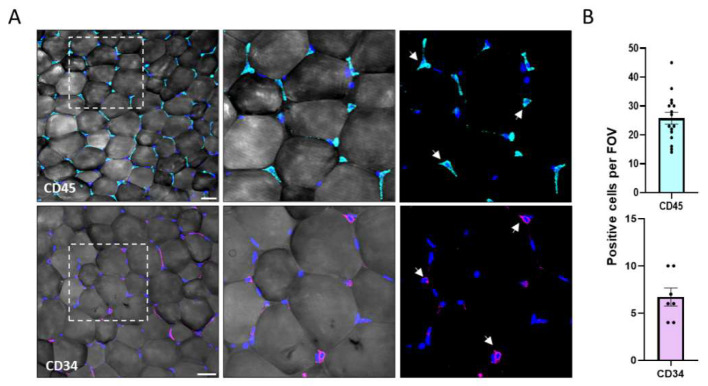
Whole-mount 3D staining of non-adipocyte subpopulations. (**A**) Immunofluorescence staining of CD45+ (cyan), CD34 +(magenta), and cell nuclei (DAPI, blue) in whole mount adipose tissue. (Magnification of ×200, scale bar = 50 μm). (**B**) Quantification of CD45+ (cyan) and CD34 + (magenta) per FOV.

**Figure 4 biomolecules-11-01906-f004:**
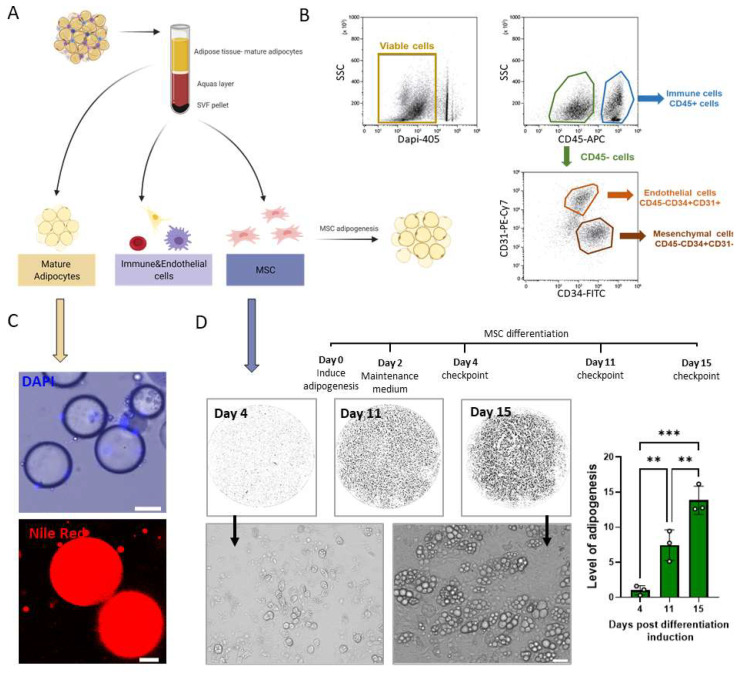
Cell subpopulation derived from adipose tissue; isolation, identification, and differentiation (**A**) Schematic illustration of the different cellular fractions isolated from adipose tissue (**B**) Flow cytometry analysis of stromal vascular fraction cells. Suspended stromal vascular cells stained with CD45-APC, CD31-PECy7, and CD34-FITC to separate immune, mesenchymal progenitors, and endothelial cell populations (Dead cells were excluded with DAPI). (**C**) Images of isolated mature adipocytes stained for lipid droplets with Nile red (scale bar = 20 μm) and nuclei with DAPI (scale bar = 50 μm). (**D**) Binary images of the adipocyte differentiation representing their LOA at 4, 11, and 15 days. Enlargement of the cells at ×200 (scale bar = 50 μm) after four and fifteen days, and the LOA analysis (n = 3), relative to day 4; significance was calculated using one-way ANOVA with Tukey’s post-test. ** *p* < 0.01 and *** *p* < 0.001. Error bars represent means ± SD.

**Figure 5 biomolecules-11-01906-f005:**
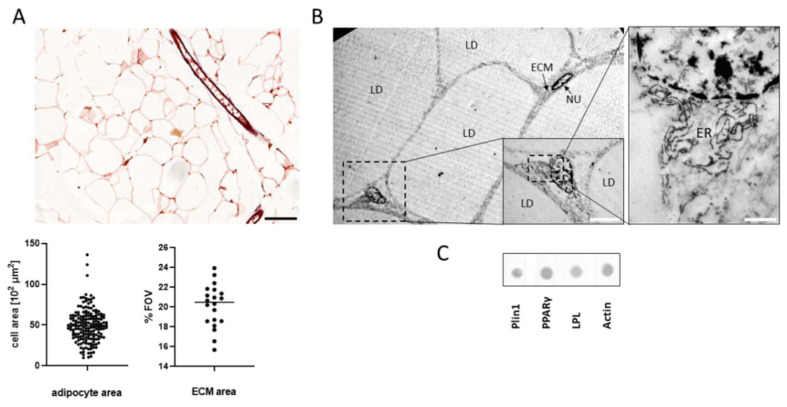
Tissue morphology and protein expression. (**A**). Histology analysis of adipose tissue (scale bar = 100 μm) used to analyze single-cell (n = 390) of adipocyte size and ECM area in FOV (n = 20) of the adipose tissue. (**B**). TEM images of adipocytes and a fibroblast were visualized. (LD-lipid droplet; ECM-extracellular matrix; NU- nucleus; ER- endoplasmic reticulum, Magnification: ×20,000 (**right**), scale bar = 1 µm; ×5000 (**left**), scale bar = 5 µm. (**C**). Immunoblotting of adipose tissue proteins shown by Dot-blot.

**Figure 6 biomolecules-11-01906-f006:**
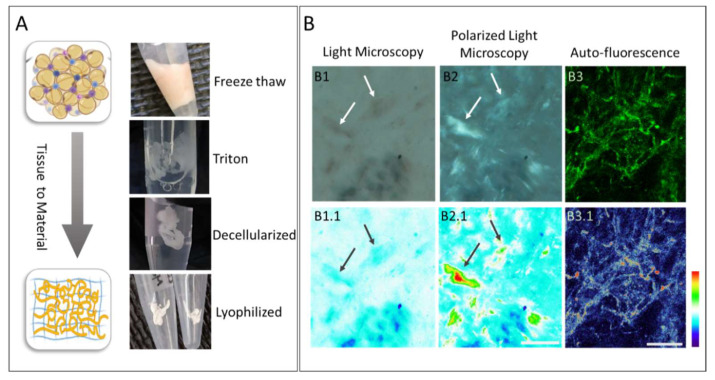
De-cellularization procedure from tissue to material and microscopy observation: (**A**) From top to bottom: WAT tissue in native state, triton treated loss in volume; tissue remaining ECM, lyophilized ECM (**B**). Microscopy images of unstained ECM (upper panel) and thermal illustration (lower panel) B1, ECM in white light, no clear areas of interest; B2, under light polarization microscope, clear areas of collagen due to birefringence properties B3, auto fluorescence of ECM accentuates fibrous structure. Color analysis Thermal depiction B1.1 clear areas shown, B2.1 areas of interest intensity comparable, B3.1 accentuates thicker areas, (B1–B2 scale bar = 100 µm, B3 scale bar = 20 µm).

## Data Availability

Data are available from the authors upon reasonable request.

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
