# Peer review of "Exploring the Cell Stemness and the Complexity of the Adipose Tissue Niche"

_biomolecules, 2021, doi:10.3390/biom11121906_

Round 1

Reviewer 1 Report

In this article is missing CD analysis for CD105, CD90 and CD73 to better characterize the mesenchymal stem cells derived from ADIPOSE. If you do not have these results, you can discuss.

Reviewer 2 Report

Kislev et al have carried out an exhaustive study in visceral adipose tissue samples to characterize the complexity of the adipose tissue.

The manuscript is well written, structured, and discussed. However, there are important points that need to be clarified.

It is difficult to see the biological relevance of these findings. The objective of the work is not clear. It seems a summation of different techniques to describe a protocol to be applied in the study of adipose tissue, but without a biological objective to answer. For example, it could apply these techniques to compare the cd34 + percentage in a control or obesity situation, or the differentiation capacity between males and females, or depending on age, to compare the cell area with TEM or microscopy, SAT vs. VAT, BAT vs. WAT, ...

In addition, although the methodology used is robust, it is necessary to explain several points.

For instance. Origin of adipose tissue. Although it was explained that visceral adipose tissue was obtained from mice, it is necessary to specify which visceral adipose tissue it is (mesenteric, gonadal, retroperitoneal,…). In addition, it is of great relevance to explain the characteristics of the mice: male / female, number of mice and housing conditions, age, approval of the ethics committee, total weight and percentage of adipose tissue, euthanasia method ...

Furthermore, it is necessary to explain the culture method to induce the differentiation of SVF in mature adipocytes (cocktails, ...) and what were the parameters analyzed to ensure the level of adipogenesis. (LOA).

Please explain the reason for separating the protein samples on SDS-PAGE gel in a Dot Blot. Besides, it is necessary to show the amount of protein loaded.

In the histological analysis, provide the software used for the area analysis calculation. In the figure 6, it is assumed that the fibrous structures were stained with Masson's Trichrome.

Minor points

Line 16. It is strange to underline the word “Abstract” and to write in bold the first word, “Adipose”.

Line 144 should be written in bold and underlined.

Reviewer 3 Report

Please critically add in the introduction the influence of the local microenvironment, especially immunological and inflammatory signals on the cell populations in the adipose tissue (https://pubmed.ncbi.nlm.nih.gov/34490235/).  Also cover the activation of BAT and beige adipose tissue in response to physiological cues (https://pubmed.ncbi.nlm.nih.gov/34625737/).

It will be also interesting to know how to expand the in vitro data to the in vivo situation ?  Do the authors have markers that can be found in vivo ?

Following the paper of Hélène Busser (https://pubmed.ncbi.nlm.nih.gov/26086188/), why the authors have not assessed the different markers proposed in the literature to identify subpopulation of MSCs within the adipose tissue ? In particular when these subpopulations harbor significant functional properties? 

The contribution of Dpp4+ progenitors to de novo adipogenesis should be also mentioned (https://pubmed.ncbi.nlm.nih.gov/34662714/).

Round 2

Reviewer 2 Report

The data presented in this updated version is much clearer compared to the last version. The results of this manuscript will be of interest to the readers. The publication of this manuscript in the current version is recommended. 

Reviewer 3 Report

Despite my recommendation, the authors have not discussed my comments and have not cited the relevant paper proposed. Please revise accordingly.